# Metformin for knee osteoarthritis with obesity: study protocol for a randomised, double-blind, placebo-controlled trial

Yuan Z Lim ![ORCID], Yuanyuan Wang ![ORCID], Donna M. Urquhart,
Mahnuma Mahfuz Estee ![ORCID], Anita E Wluka, Stephane Heritier, Flavia M Cicuttini ![ORCID]

YZL and YW contributed equally.

School of Public Health and Preventive Medicine, Monash University, Melbourne, Victoria 3004, Australia

**Correspondence to**
Professor Flavia M Cicuttini;
flavia.cicuttini@monash.edu

## ABSTRACT

**Introduction** Over half of the populations with knee osteoarthritis (OA) have obesity. These individuals have many other shared metabolic risk factors. Metformin is a safe, inexpensive, well-tolerated drug that has pleiotropic effects, including structural protection, anti-inflammatory and analgesic effects in OA, specifically the knee. The aim of this randomised, double-blind, placebo-controlled trial is to determine whether metformin reduces knee pain over 6 months in individuals with symptomatic knee OA who are overweight or obese.

**Methods and analysis** One hundred and two participants with symptomatic knee OA and overweight or obesity will be recruited from the community in Melbourne, Australia, and randomly allocated in a 1:1 ratio to receive either metformin 2 g or identical placebo daily for 6 months. The primary outcome is reduction of knee pain [assessed by 100 mm Visual Analogue Scale (VAS)] at 6 months. The secondary outcomes are OMERACT-OARSI (Outcome Measures in Rheumatology-Osteoarthritis Research Society International) responder criteria [Western Ontario and McMaster Universities Osteoarthritis Index (WOMAC) pain, function and participant's global assessment (VAS)] at 6 months; change in knee pain, stiffness, function using WOMAC at 6 months and quality of life at 6 months. Adverse events will be recorded. The primary analysis will be by intention to treat, including all participants in their randomised groups.

**Ethics and dissemination** Ethics approval has been obtained from the Alfred Hospital Ethics Committee (708/20) and Monash University Human Research Ethics Committee (28498). Written informed consent will be obtained from all the participants. The findings will be disseminated through peer-review publications and conference presentations.

**Trial registration number** ACTRN12621000710820 .

## STRENGTHS AND LIMITATIONS OF THIS STUDY

⇒ This study is a randomised, double-blind, placebo-controlled trial.
⇒ This study will provide high-quality evidence to address whether metformin has an analgesic effect over 6 months in individuals with symptomatic knee osteoarthritis with overweight or obesity.
⇒ The generalisability of this study result will be limited to those without diabetes or those not requiring antihyperglycaemic therapy.

a multifactorial disease, management of OA has taken a 'one-size-fits-all' approach without considering the different pathological pathways and OA phenotypes, resulting in poor patient outcomes. One distinctive knee OA phenotype is the obese phenotype,[4 5] mediated by inflammatory and metabolic mechanisms.[6] Over 50% of knee OA patients have obesity.[7] Given obesity and obesity-related metabolic factors (hyperglycaemia, dyslipidaemia, hypertension) are all risk factors for knee OA,[6 8] drugs targeting obesity and its associated inflammatory and metabolic abnormalities have the potential to affect the pathogenesis of knee OA.

Metformin is a safe, inexpensive, well-tolerated oral biguanide, which is not only widely used for treatment of type 2 diabetes for over 60 years, but also has a long history of safe use in non-diabetic populations.[9 10] Additional to its glucose lowering effects, metformin modulates metabolic factors, resulting in at least 2–3 kg of weight loss[11 12] and reduced inflammation and plasma lipids.[9 10] A recent systematic review of animal and human studies showed metformin has structural protective, anti-inflammatory and analgesic effects for OA, specifically for the knee.[13] These pleiotropic effects of metformin are mainly driven by the activation of AMP-activated protein kinase (AMPK) pathway.[14–16] Hence, metformin has the potential to reduce pain in those with

## INTRODUCTION

Osteoarthritis (OA) is a leading cause of global disability, resulting in 19 million years lived with disability in 2019.[1] There is no approved disease-modifying treatment for OA to date. With limited effective therapies, end-stage OA is treated with total joint replacement, estimated to cost about US$10 billion/year in the USA[2] and over $A1 billion/year in Australia.[3] Despite being

knee OA and overweight or obesity. This study aims to determine the effect of metformin on reducing knee pain in people with symptomatic knee OA and overweight or obesity.

## METHODS AND ANALYSIS

### Study design

This is a randomised, double-blind, placebo-controlled trial in people with symptomatic knee OA and overweight or obesity, to determine the effect of metformin 2 g daily versus placebo on reducing knee pain over 6 months.

### Hypothesis and objectives

It is hypothesised that metformin, compared with placebo, will (1) reduce knee pain (primary hypothesis); (2) improve clinical outcomes (stiffness, function and health-related quality of life) and that (3) the effect of metformin on knee pain and function will be associated with changes in inflammatory and metabolic biomarkers and/or weight loss. If metformin is proven to be effective, it will provide a safe, low-cost treatment to reduce pain and improve function for people with symptomatic knee OA with concurrent overweight or obesity.

### Trial registration and reporting

The trial was registered at the Australian New Zealand Clinical Trials Registry prior to commencing recruitment (ACTRN12621000710820, registered 8 June 2021). The trial reporting will be guided by the Consolidated Standards of Reporting Trials Statement.[17]

### Study setting and participants

Participants with symptomatic knee OA and overweight or obesity will be recruited using a combined strategy including collaboration with medical practitioners and advertisements in social and local media. This single-centre study will be conducted in Melbourne, Australia.

### Inclusion criteria

Men and women aged >40 years, with overweight or obesity (body mass index $\geq 25 \, \text{kg/m}^2$); (2) Knee pain for at least 6 months with a pain score >40 mm on a 100 mm Visual Analogue Scale (VAS) and (3) Meet the American College of Rheumatology clinical criteria for knee OA.[18]

### Exclusion criteria

(1) Severe radiographic knee OA (Kellgren-Lawrence grade 4) or severe knee pain (on standing >80 mm on a 100 mm VAS); (2) Any inflammatory arthritis including rheumatoid arthritis, psoriatic arthritis, crystal arthritis, spondyloarthritis, connective-tissue disease associated arthritis or reactive arthritis or significant knee injury; (3) Known or newly diagnosed diabetes requiring anti-hyperglycaemic therapy or previous adverse reaction to metformin; (4) Index knee surgery (arthroscopy or open surgery) in the past year; (5) Index knee intra-articular hyaluronic acid injection in the past 6 months or corticosteroid injection in the past 3 months; (6) Use of any investigational drugs or device within 30 days prior to randomisation; (7) Index knee planned joint replacement or arthroscopy in the next 6 months; (8) Other muscular, joint or neurological condition affecting lower limb function; (9) Acute or chronic renal or liver impairment; (10) Other medical condition precluding study participation or relocation and (11) Women who are pregnant, lactating or trying to become pregnant. Use of menopausal hormone therapy or contraceptive pill will be permitted so long as the dose has been stable for at least 30 days prior to study entry.

### Study timeline

This trial began recruitment on 16 June 2021. It is estimated to finish recruitment on 30 September 2023 and complete the 6-month follow-up and data collection in March 2024. Figure 1 shows trial participation and study procedure.

### Randomisation, allocation concealment and blinding

Allocation of participants in a 1:1 ratio to one of the two groups will be by computer-generated random numbers prepared by a statistician with no involvement in the trial. Block randomisation using random permuted blocks of sizes 4 and 6 will be performed. The use of a central automated allocation procedure with security in place will ensure the allocation cannot be accessed or influenced by any person. Allocation concealment and double blinding will be ensured by: (1) medications being dispensed by Syntro Pharmacy; (2) use of an identical placebo tablet and (3) subjective measures being taken by research assistants blinded to group allocation. Participants, assessors and statisticians will be blinded to group allocation.

### Intervention and dosing

All participants will undergo usual care by their treating health practitioners. Eligible participants will be randomly assigned to receive either metformin (up to 2000 mg) once daily or placebo once daily. Treatment with study drug will be initiated at a dose of 500 mg once a day with the evening meal. Over 6 weeks, the dose of study drug will be titrated to 2000 mg once daily (or placebo once daily) to minimise gastrointestinal side effects.

### Safety

Any adverse events or serious adverse event will be reported during the study. Blood tests will occur at screening. No data safety monitoring board is required as this agent is approved by Therapeutic Goods Administration with a well-known safety profile.[19] Unblinding participants due to side effects of metformin was not an issue in a previous clinical trial.[12]

### Compliance

Compliance with trial medication will be assessed at 6 months by pill count. Study staff will phone participants in the middle between study visits to monitor medication adherence. Monthly telephone contact for the first 5 months will be conducted to address any concerns, as

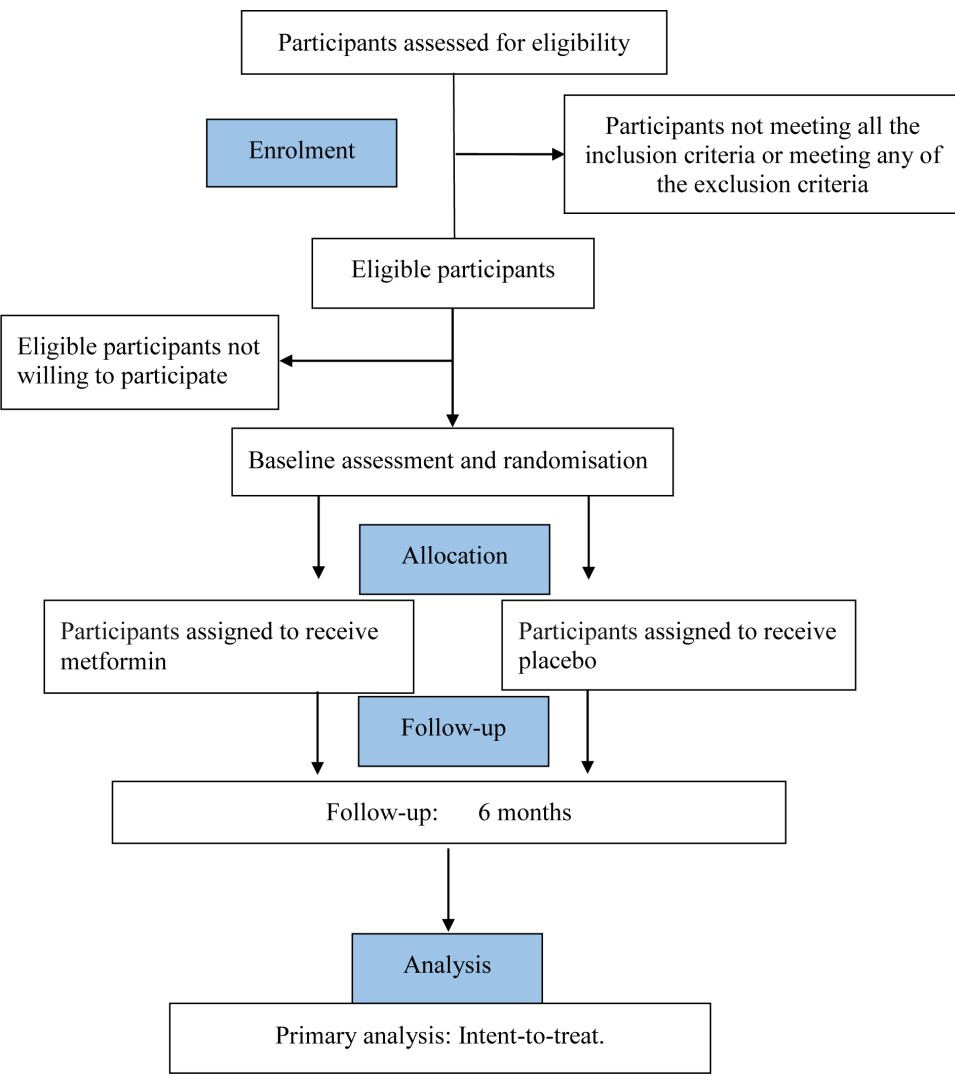

**Figure 1** Flow chart of trial participation.

well as following up knee pain outcome (VAS). This will help to mitigate non-compliance.

### Concomitant medication
To maintain the pragmatic nature of the trial, there are no restrictions with regard to concomitant medications, including glucosamine, chondroitin, non-steroidal anti-inflammatory drugs and opioids analgesics, which will be allowed during the trial and be recorded by questionnaire at all visits. Patients will be asked to keep medications as stable as possible and use paracetamol as rescue medication.

### Study procedures
Participants will be screened via phone by questionnaire before attending the screening visit via telehealth. There will be two study visits (onsite or telehealth): screening/baseline and month 6, as shown in table 1. At screening, participants will complete questionnaires, have a knee X-ray and blood tests (renal and liver function, plasma glucose and lipids, vitamin $B_{12}$ and inflammatory biomarkers (C reactive protein, tumour necrosis factor,

interleukin-1 and interleukin 6)) to ensure inclusion criteria are met and exclusion criteria are absent. The index knee will be defined as having symptomatic OA. If both knees are symptomatic and eligible based on VAS pain, the one with higher VAS pain score will be used; if both knees are symptomatic with the same pain level, the one with least severe radiographic OA (joint space narrowing) will be used; if both knees have the same pain level and radiographic severity of OA, the dominant knee will be used. Physical examinations and questionnaires will be performed at months 0 and 6. The same researchers, blinded to treatment allocation, will measure all clinical variables, administer questionnaires, monitor compliance and record adverse events. Participants are able to withdraw at any time during the trial; the time and reasons will be recorded. If participants withdraw from the study, they will be requested to complete questionnaires (posted to the participants with a return envelope).

### Primary outcome
The primary outcome is pain reduction at 6 months, measured by change in VAS knee pain from baseline to

**Table 1** Schedule of assessments

|  | Screening | Double-blind period | | |
|  | Screening/ baseline assessment | Randomisation | 1–5 months | 6 months |
| --- | --- | --- | --- | --- |
| Study visit (onsite or telehealth) | X |  |  | X |
| Telephone interview (monthly) |  |  | X |  |
| Informed consent | X |  |  |  |
| Knee X-ray | X |  |  |  |
| Blood test | X |  |  |  |
| Medical history/conditions | X |  |  | X |
| Medication | X |  |  | X |
| Employment and education | X |  |  |  |
| Smoking and alcohol | X |  |  |  |
| Questionnaires |  |  |  |  |
| Knee VAS | X |  | X | X |
| WOMAC | X |  |  | X |
| PainDETECT | X |  |  | X |
| Hand VAS | X |  | X | X |
| Multisite pain | X |  |  | X |
| AQoL | X |  |  | X |
| IPAQ | X |  |  | X |
| Physical examination |  |  |  |  |
| Height, weight* | X |  |  | X |
| Compliance and safety (adverse events) |  |  | X | X |
| Dispense medication |  | X |  |  |

*Height and weight will be self-reported if the visit is via telehealth.
AQoL, assessment of quality of life; IPAQ, international physical activity questionnaire; VAS, Visual Analogue Scale; WOMAC, Western Ontario and McMaster Universities Osteoarthritis Index.

6 months (follow-up VAS pain score—baseline score). Knee pain will be measured at baseline and monthly follow-up using a 100 mm VAS by asking 'on this line, where would you rate your knee pain over the last 7 days?' with terminal descriptors 'no pain' (score 0) and 'worst imaginable pain' (score 100).

### Secondary outcomes
#### OMERACT-OARSI responder criteria
This will be used to define a responder based on improvement in Western Ontario and McMaster Universities Osteoarthritis Index (WOMAC) pain and function and the participant's global assessment[20] at 6 months. Participant's global assessment will be evaluated by 100 mm VAS.[21]

#### Change in knee pain, stiffness and function
Knee pain, stiffness and function will be assessed using the WOMAC[22] at baseline and 6 months.

#### Health-related quality of life
This will be measured using the Assessment of Quality of Life[23] at baseline and 6 months.

### Other measures
#### Descriptive data
Data regarding age, gender, height, weight, duration of symptoms, employment, medical history, medication use, education level, smoking, alcohol consumption will be collected using a questionnaire at baseline.

#### PainDETECT
PainDETECT is a validated questionnaire used to assess pain sensitisation in OA[24] and will be assessed at baseline and 6 months.

#### Physical activity
Physical activity will be measured using the International Physical Activity Questionnaire[25] short version at baseline and 6 months.

#### Hand VAS
Pain reduction in hands will be measured at baseline, then monthly for 6 months, according to the OARSI recommendations for the design and conduct of clinical trials for hand OA, which recommend the use of single question pain VAS.[26]

## Multisite pain

The presence and levels of multisite musculoskeletal pain will be assessed at baseline and 6 months using a questionnaire.

## Adverse events

These will be measured in a log-book by the blinded assessor at each follow-up.

## Biochemical parameters

General (cell counts, liver and renal function), inflammatory biomarkers (C reactive protein, erythrocyte sedimentation rate, interleukin-6, tumour necrosis factor), plasma glucose and lipids, and vitamin $B_{12}$ will be measured at baseline and 6 months.

## Knee X-ray

X-ray of the study knee (weight-bearing anteroposterior view) will be scored using Kellgren-Lawrence grade. Our intraobserver and interobserver reliability is 0.93 and 0.86 for osteophytes, 0.93 and 0.85 for joint space narrowing, respectively.[27]

## Sample size calculation

The primary outcome is change in VAS knee pain over 6 months. The mean VAS pain was 55 mm (out of 100 mm) in our previous knee OA clinical trial with similar eligibility criteria.[28 29] Using the control group data, we assume a between-participant SD of change in VAS pain of 24 mm. With 41 participants per arm, the study will have 80% power to detect a 15 mm difference in VAS pain between the intervention and control groups which is the minimum clinically important difference to be detected in OA trials,[21] alpha 0.05, two-sided significance. Based on our previous knee OA trials,[28 30–32] with a conservative assumption of 20% lost to follow-up, we will recruit 102 participants (51 in each arm of the study).

## Statistical analyses

Intention-to-treat analyses of primary and secondary outcomes will be presented, including all participants in their randomised groups. Comparisons between randomised groups of change in knee pain at 6 months will be analysed using analysis of covariance (ANCOVA), adjusting for baseline value for knee pain outcomes. Differences in knee pain trajectories over 6 months will be examined using linear mixed-effects models with baseline value as the covariate, fixed factors for treatment, time and treatment×time interaction, and with an autoregressive (1) covariance matrix for repeated measures of individuals over time. Sensitivity analyses will be conducted for clinically important imbalances in baseline factors using multiple linear regression, or mixed models regression, as appropriate for the outcome measures. Multiple imputation of missing follow-up measures will be carried out as a sensitivity analysis when the percentage of missing data exceeds 5%. Subgroup analyses will be performed to examine whether the difference in outcomes between randomised groups varied based on sex, knee pain level and radiographic severity of knee OA. Analyses of treatment efficacy will be done by blinding individuals at the time of any protocol deviation and developing a model for the probability of deviation, followed by analyses using only the uncensored individuals where the weights are the inverse probability of censoring. This produces estimates of treatment effect as if there was full compliance with the protocol in this randomised controlled trial and is far preferable to per-protocol analyses based on (unweighted) observed compliance.[33]

## Data integrity and management

All data will be collected using Monash Research Electronic Data Capture (REDCap). Paper copies of questionnaires (if participants prefer to complete the questionnaires on hard copy) will be stored in locked filing cabinets, with restricted access. Electronic data will be stored in REDCap, and exported to a password-protected server after data collection, separating the identifying and non-identifying information. The codes linking data to identifying participant information will be kept separately from the study data, under password protection and with restricted access.

Due to the COVID-19, we will be providing a telehealth option for all clinic visits. This will be done in such a way that will not compromise participant safety or the scientific integrity of the trial. This study uses REDCap for consent and data collection, facilitating telehealth options. For participants who use the telehealth option for the screening/baseline visit, we will seek consent electronically (eConsent). REDCap has a feature that implements consent forms through an online survey which can be accessed on a computer, mobile phone or tablet. The completed eConsent portable document format (PDF) s are stored in REDCap in a file repository under 'PDF Survey Archive'. Physical examination will not be possible with telehealth option. Thus, height and weight will be self-reported.

## Patient and public involvement

This study was informed by identification of clinical need in patients with OA attending our clinics. The clinical need and approach to translation is informed by the work with Musculoskeletal Australia in systematic reviews of consumers' needs in OA.[34] Once the trial has been published, participants will receive a study newsletter with details of the results which is suitable for a non-specialist audience.

## Ethics and dissemination

Ethics approval has been obtained from the Alfred Hospital Ethics Committee (708/20) and Monash University Human Research Ethics Committee (28498). Written informed consent will be obtained from all the participants (online supplemental document). Trial results, regardless of statistical significance, will be published in peer-reviewed journals and presented at national and international conferences. On publication of the primary

manuscript, participants will be informed of their group allocation and provided with the results.

## DISCUSSION

This randomised controlled trial is conducted to determine whether metformin 2 g daily over 6 months reduces knee pain in participants with symptomatic knee OA and concurrent overweight or obesity. If metformin proves effective in patients with symptomatic knee OA and concurrent overweight or obesity, it will offer an important therapeutic approach for obesity-metabolic syndrome phenotype of knee OA.

There are consistent chondroprotective, immunomodulatory and analgesic effects from metformin in preclinical cell and animal studies.[13] In preclinical studies, in addition to chondroprotective and anti-inflammatory effects, metformin was shown to be able to improve pain, such that rats or mice treated with metformin showed increased paw withdraw latency indicative of reduction in pain.[13 15 16 35] In human studies, a randomised, double-blind trial showed that the combination of metformin with meloxicam improved knee pain by at least 50% more than meloxicam alone.[36] Additionally, one of the metformin's pleiotropic effects is mild weight loss (~2.5%),[37] which is important when tackling the slow insidious weight creep from early to middle adulthood,[38–40] particularly when obesity is a well-known risk factor for OA, and for more symptomatic and more progressive knee OA. Slowing weight gain over time not only has been proven to improve knee pain,[41] but also was estimated to reduce knee replacement by up to 28.4%.[42] As such, metformin has the potential to play an important role in individuals who have knee OA with obesity-metabolic syndrome phenotype.

Studies have shown the beneficial effects of metformin in OA were mainly mediated by activation of the AMPK pathway.[13] As a key regulator of energy homeostasis and metabolism, activation of AMPK regulates key downstream enzymes involved in metabolism and transcription factors that regulate gene expression. As such, activation of the AMPK pathway in liver, muscle and adipose results in decreased lipogenesis and increased fatty acid oxidation, explaining some of the pleiotropic effects of metformin in improving metabolic profiles.[10]

The study has several strengths. It is a randomised, double-blind, placebo-controlled trial which will provide high-quality evidence to address the aim of this study. Nevertheless, our study population is limited to those without a valid indication for metformin use, as it would be unethical to withhold metformin with a clinical indication, specifically people with diabetes, thus limiting the generalisability of the study results. The diabetic population is known to have more obesity and concurrent metabolic syndrome,[43] and it is likely that those with diabetes and knee OA who will be excluded from this study, are the populations at greatest need for metformin, which may underestimate the potential effect of metformin in this study.

In summary, knee OA, specifically the obesity-metabolic syndrome phenotype, has limited effective treatment options. This study will provide high-quality evidence to determine whether metformin reduces knee pain in people with symptomatic knee OA with overweight or obesity over 6 months, with major clinical and public health importance for a potentially effective treatment option for knee OA to reduce knee pain and disease burden.

**Acknowledgements** Ms Molly Bond, Mr Noor Abid, Mr Ashish Dinesh Nair, Dr Benjamin Sutu, Dr Talia Igel, Dr Luigi Zolio and Dr Rushab Shah have been involved in the coordination and/or execution of this study.

**Contributors** Conception and design: YW, DMU, AW and FC. Study execution and data acquisition: YL, YW, MME, AW and FC. Drafting of the manuscript: YL, YW and FC. Critical revision of the manuscript for important intellectual content and approval of the final manuscript: YW, DMU, MME, AW, SH and FC. Obtaining of funding: FC.

**Funding** The study is funded by an Investigator Grant from the National Health and Medical Research Council of Australia (NHMRC APP1194829). YL is the recipient of National Health and Medical Research Council (NHMRC) Clinical Postgraduate Scholarship (#1133903) and Royal Australasian College of Physicians Woolcock Scholarship. YW is the recipient of NHMRC Translating Research into Practice Fellowship (APP1168185). DMU is a recipient of an NHMRC/Medical Research Future Fund (MRFF) Career Development Fellowship (Clinical Level 2 #1142809). MME is the recipient of Bangabandhu Science and Technology Fellowship from Ministry of Science and Technology, Government of the People's Republic of Bangladesh. FC is the recipient of NHMRC Investigator Grant (APP1194829).

**Disclaimer** The funders of the study had no role in the study design and conduct of the study; collection, management, analysis and interpretation of the data; preparation, review, or approval of the manuscript; and decision to submit the manuscript for publication.

**Competing interests** None declared.

**Patient and public involvement** Patients and/or the public were involved in the design, or conduct, or reporting, or dissemination plans of this research. Refer to the Methods section for further details.

**Patient consent for publication** Not applicable.

**Provenance and peer review** Not commissioned; externally peer reviewed.

**ORCID iDs**
Yuan Z Lim http://orcid.org/0000-0001-6960-8813
Yuanyuan Wang http://orcid.org/0000-0002-7629-4178
Mahnuma Mahfuz Estee http://orcid.org/0000-0002-4714-7195
Flavia M Cicuttini http://orcid.org/0000-0002-8200-1618

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
