## [Reviewer comments · BMJ Open]

ARTICLE DETAILS

TITLE (PROVISIONAL)	Metformin for knee osteoarthritis with obesity: study protocol for a randomised, double-blind, placebo-controlled trial
AUTHORS	Lim, Yuan; Wang, Yuanyuan; Urquhart, Donna M.; Estee, Mahnuma; Wluka, Anita; Heritier, Stephane; Cicuttini, Flavia

VERSION 1 – REVIEW

REVIEWER	Nadjarzadeh, Azadeh Shahid Sadoughi University of Medical Sciences and Health Services
REVIEW RETURNED	18-Sep-2023

GENERAL COMMENTS	The manuscript has been written very carefully.
---

REVIEWER	Maheu, E Hopital Saint-Antoine
REVIEW RETURNED	25-Oct-2023

GENERAL COMMENTS	This is a paper reporting on a protocol aiming at studying the symptomatic, primarily analgesic, effect of metformin in individuals with Overweight/obesity associated knee OA. Indeed, this is a very enthusiastic field of research since we do not have today any OA-targeted treatment, and since it has become obvious that new treatments should target and/or take into account the different OA phenotypes. Among them, the metabolic phenotype is now very much studied and the relationship between glucose metabolism, fatty acids metabolism, inflammation and OA is currently established. This paper reports on the protocol of an ongoing RCT studying metformin at a 2000 mg/ day dosage versus placebo. The trial recruitment started June 16, 2021 and is expected to be completed end of September. Therefore, my comments will not focus on eventual / suggested modification of this protocol. Overall, this is a very exciting promising and well-designed study which results are impatiently expected. Page 5: Ethics approval: You should provide the consent form for written approval given to the patients enrolment in the appendix.
---

	Page 6: Exclusion: line 2: you should maybe specify what you included in “inflammatory arthritis”? Crystal-induced arthritis? Hemophilic arthritis? post-septic arthritis recurrent flares? Page 6: regarding pregnancy, did you asked women to be under contraception? Page 6: randomization: Why did you use 2 different sizes of blocks (4 and 6)? Page 7: intervention: Could you specify how was planned the increase of the daily dosage of metformin from 500 mg to 2000 mg over 6 weeks? Which steps were used? Page 7: concomitant medication: What about other non-OA concomitant medications? Were they monitored (i.e.... statin intake for instance)? Page 7: study procedure: Why did you not maintain an initial baseline physical visit? How, therefore, if telehealth, did you assess physical parameters, such as weight and height with accuracy? Page 8: study procedures continuing: did you plan a physical examination, other than weight and height and PROs? For instance, and this may be of importance for the results, did you look at eventual effusion, flossum, lateral knee deviation, previous history of surgery in the index knee before the past year (this OA could be traumatic OA after cruciate anterior ligament and/or meniscus injury)? Dominant knee: how did you plan the assessment of knee dominance (although the condition in which such an assessment was required might have been infrequent)? Page 10: end of 2nd line, can you provide the reference of your previous knee OA trial from which you used the numbers to calculate your sample size? Following line, you state that a 15 mm difference on a pain VAS is the MCID for clinical trials. This is quite a high value, many papers stating that rather that the MCID is around 10-11 mm. Page 10: Statistics: considering potential subgroup analyses, I would suggest adding an exploratory analysis on the type of recruitment (advertisement vs doctors). Page 11, end of the 1st 2 paragraphs, again, what about the physical examination, e.g. my previous comment on page 8? Page 12: discussion: when you write “particularly when obesity is a well-known risk factor for OA”, I would specify and add “for OA, and for more symptomatic and more progressive knee OA”. Page 15: flowchart: In the box “analysis”, I would maybe specify “Intent-to-treat” or “primary: intent-to-treat”.
--	--

	Page 16: Table 1: Height, weight: how are they accurately measured in case of a telehealth visit? Did you record other parameters: effusion, flossum, deviation, etc.?
--	--

VERSION 1 – AUTHOR RESPONSE

Reviewer: 2

Dr. E Maheu, Hopital Saint-Antoine, St Antoine Hospital

Comments to the Author:

This is a paper reporting on a protocol aiming at studying the symptomatic, primarily analgesic, effect of metformin in individuals with Overweight/obesity associated knee OA.

Indeed, this is a very enthusiastic field of research since we do not have today any OA-targeted treatment, and since it has become obvious that new treatments should target and/or take into account the different OA phenotypes. Among them, the metabolic phenotype is now very much studied and the relationship between glucose metabolism, fatty acids metabolism, inflammation and OA is currently established.

This paper reports on the protocol of an ongoing RCT studying metformin at a 2000 mg/ day dosage versus placebo. The trial recruitment started June 16, 2021 and is expected to be completed end of September.

Therefore, my comments will not focus on eventual / suggested modification of this protocol.

Overall, this is a very exciting promising and well-designed study which results are impatiently expected.

Response:

Thank you for your kind comments.

Page 5: Ethics approval: You should provide the consent form for written approval given to the patients enrolment in the appendix.

Response:

We have now provided the consent form as a supplemental document in the appendix.

Page 6: Exclusion: line 2: you should maybe specify what you included in “inflammatory arthritis”? Crystal-induced arthritis? Hemophilic arthritis? post-septic arthritis recurrent flares?

Response:

We have specified the types of inflammatory arthritis excluded for this study.

Page 6

Any inflammatory arthritis including rheumatoid arthritis, psoriatic arthritis, crystal arthritis, spondyloarthritis, connective-tissue disease associated arthritis or reactive arthritis or significant knee injury.

Page 6: regarding pregnancy, did you asked women to be under contraception?

Response:

We did not ask women participants to be under contraception.

Metformin is listed as a category C drug in Australia (i.e. Drugs which, owing to their pharmacological effects, have caused or may be suspected of causing, harmful effects on the human fetus or neonate without causing malformations. These effects may be reversible). While it is not formally approved for use in pregnancy, maternal use of metformin has not been associated with an increased risk of congenital malformations or adverse pregnancy outcomes[1-4]. Therefore, we did not specially request women participants to be under contraception.

Page 6: randomization: Why did you use 2 different sizes of blocks (4 and 6)?

Response:

We used permuted blocks of varying size (4 and 6) for randomisation. We have amended the sentence.

Page 6:

Block randomisation using random permuted blocks of sizes 4 and 6 will be performed.

Page 7: intervention: Could you specify how was planned the increase of the daily dosage of metformin from 500 mg to 2000 mg over 6 weeks? Which steps were used?

Response:

The planned increase of metformin dosage is detailed in the participant information sheet provided to all participants. This has now been provided as a supplemental material (Participant information sheet/consent form). The planned increase of dosage aims to limit potential gastrointestinal side effects of metformin.

Week 1-2: 500mg (1 x 500mg tablet) per day

Week 3-4: 1000mg (2 x 500mg tablets) per day

Week 5-6: 1500mg (3 x 500mg tablets) per day

Week 7-8 and onwards: 2000mg (2 x 1000mg tablets) per day

Page 7: concomitant medication: What about other non-OA concomitant medications? Were they monitored (i.e.... statin intake for instance)?

Response:

All concomitant medications, including non-OA concomitant medications, are documented at baseline study visit and at 6 months follow up; analgesics use is documented at each monthly follow ups.

Page 7: study procedure: Why did you not maintain an initial baseline physical visit? How, therefore, if telehealth, did you assess physical parameters, such as weight and height with accuracy?

Response:

This study began recruitment in June 2021, where Melbourne was still under many strict Covid pandemic restrictions, including the multiple lockdowns. The Covid pandemic restrictions were only removed in September 2022. Therefore, to allow ongoing recruitment, telehealth became the safe, preferred method of choice for baseline visit. As such, self-reported weight and height were obtained.

Page 8: study procedures continuing: did you plan a physical examination, other than weight and height and PROs?

Response:

We did not plan a physical examination for any other patient reported outcomes apart from weight and height measurements.

For instance, and this may be of importance for the results, did you look at eventual effusion, flossum, lateral knee deviation, previous history of surgery in the index knee before the past year (this OA could be traumatic OA after cruciate anterior ligament and/or meniscus

injury)?

Response:

We did not look at knee effusion but as part of the medical history any previous injury, surgery or procedure done to the study knee will be recorded. However, all participants will have a knee X-ray as part of screening.

Dominant knee: how did you plan the assessment of knee dominance (although the condition in which such an assessment was required might have been infrequent)?

Response:

The dominant knee is defined as the leg that participant uses to kick a ball.

Page 10: end of 2nd line, can you provide the reference of your previous knee OA trial from which you used the numbers to calculate your sample size?

Response:

Thank you for your comments. We have referenced it as below.

Page 10-11

Based on our previous knee OA trials[29, 31-33], with a conservative assumption of 20% loss to follow up, we will recruit 102 participants (51 in each arm of the study).

29. Bennell KL, Paterson KL, Metcalf BR, Duong V, Eyles J, Kasza J, Wang Y, Cicuttini F, Buchbinder R, Forbes A et al: *Effect of Intra-articular Platelet-Rich Plasma vs Placebo Injection on Pain and Medial Tibial Cartilage Volume in Patients With Knee Osteoarthritis: The RESTORE Randomized Clinical Trial. Jama 2021, 326(20):2021-2030.*

31. Cai G, Aitken D, Laslett LL, Pelletier JP, Martel-Pelletier J, Hill C, March L, Wluka AE, Wang Y, Antony B et al: *Effect of Intravenous Zoledronic Acid on Tibiofemoral Cartilage Volume Among Patients With Knee Osteoarthritis With Bone Marrow Lesions: A Randomized Clinical Trial. Jama 2020, 323(15):1456-1466.*

32. Wang Y, Jones G, Hill C, Wluka AE, Forbes AB, Tonkin A, Hussain SM, Ding C, Cicuttini FM: *Effect of atorvastatin on knee cartilage volume in patients with symptomatic knee osteoarthritis: results from a randomised placebo-controlled trial. Arthritis & Rheumatology 2021, n/a(n/a).*

33. Cai G, Jones G, Cicuttini FM, Wluka AE, Wang Y, Hill C, Keen H, Antony B, Wang X, de Graaff B et al: *Study protocol for a randomised controlled trial of diacerein versus placebo to treat knee osteoarthritis with effusion-synovitis (DICKENS). Trials 2022, 23(1):768.*

Following line, you state that a 15 mm difference on a pain VAS is the MCID for clinical trials. This is quite a high value, many papers stating that rather that the MCID is around 10-11 mm.

Response:

There are highly heterogeneous values of the minimal clinically important difference based on different calculation methods[5]. This current study and our previous trial[6] have used 15mm difference on pain VAS. The 15mm difference on pain VAS was chosen based on the minimal clinically important improvement (MCII) in clinical trials to provide meaningful information as the MCII was shown to be affected by initial degree of severity of symptoms, but not by age, disease duration or sex[7].

Page 10: Statistics: considering potential subgroup analyses, I would suggest adding an exploratory analysis on the type of recruitment (advertisement vs doctors).

Response:

Our experience from all our previous studies showed almost all Melbourne-based recruitments were obtained from social media advertisement[8, 9]. Therefore, we have not added subgroup analysis based on the type of recruitment.

Page 11, end of the 1st 2 paragraphs, again, what about the physical examination, e.g. my previous comment on page 8?

Response:

We have added a sentence to clarify that physical examination will not be possible with telehealth option.

Page 12

Physical examination will not be possible with telehealth option. Thus, height and weight will be self-reported.

Page 12: discussion: when you write “particularly when obesity is a well-known risk factor for OA”, I would specify and add “for OA, and for more symptomatic and more progressive knee OA”.

Response:

Thank you for your suggestion. We have amended the sentence accordingly.

Page 13

“... particularly when obesity is a well-known risk factor for OA, and for more symptomatic and more progressive knee OA.”

Page 15: flowchart: In the box “analysis”, I would maybe specify “Intent-to-treat” or “primary: intent-to-treat”.

Response:

We have added a box under “analysis” to state the primary analysis will be intention-to-treat analysis.

Page 16: Table 1: Height, weight: how are they accurately measured in case of a telehealth visit? Did you record other parameters: effusion, flossum, deviation, etc.?* ****

Response

We have added a note in table 1 to clarify that height and weight will be self-reported in telehealth visit.

Page 9

** height and weight will be self-reported if the visit is via telehealth.*

References

1. Abolhassani N, Winterfeld U, Kaplan YC, Jaques C, Minder Wyssmann B, Del Giovane C, Panchaud A: **Major malformations risk following early pregnancy exposure to metformin: a systematic review and meta-analysis.** *BMJ Open Diabetes Res Care* 2023, **11**(1).
2. Butalia S, Gutierrez L, Lodha A, Aitken E, Zakariasen A, Donovan L: **Short- and long-term outcomes of metformin compared with insulin alone in pregnancy: a systematic review and meta-analysis.** *Diabet Med* 2017, **34**(1):27-36.
3. Lin SF, Chang SH, Kuo CF, Lin WT, Chiou MJ, Huang YT: **Association of pregnancy outcomes in women with type 2 diabetes treated with metformin versus insulin when becoming pregnant.** *BMC Pregnancy Childbirth* 2020, **20**(1):512.
4. Feig DS, Sanchez JJ, Murphy KE, Asztalos E, Zinman B, Simmons D, Haqq AM, Fantus IG, Lipscombe L, Armson A *et al*: **Outcomes in children of women with type 2 diabetes exposed to metformin versus placebo during pregnancy (MiTy Kids): a 24-month follow-up of the MiTy randomised controlled trial.** *Lancet Diabetes Endocrinol* 2023, **11**(3):191-202.

5. Franceschini M, Boffa A, Pignotti E, Andriolo L, Zaffagnini S, Filardo G: **The Minimal Clinically Important Difference Changes Greatly Based on the Different Calculation Methods.** *Am J Sports Med* 2023, **51**(4):1067-1073.
6. Cai G, Jones G, Cicuttini FM, Wluka AE, Wang Y, Hill C, Keen H, Antony B, Wang X, de Graaff B *et al*: **Study protocol for a randomised controlled trial of diacerein versus placebo to treat knee osteoarthritis with effusion-synovitis (DICKENS).** *Trials* 2022, **23**(1):768.
7. Tubach F, Ravaud P, Baron G, Falissard B, Logeart I, Bellamy N, Bombardier C, Felson D, Hochberg M, van der Heijde D *et al*: **Evaluation of clinically relevant changes in patient reported outcomes in knee and hip osteoarthritis: the minimal clinically important improvement.** *Ann Rheum Dis* 2005, **64**(1):29-33.
8. Wang Y, Jones G, Hill C, Wluka AE, Forbes AB, Tonkin A, Hussain SM, Ding C, Cicuttini FM: **Effect of atorvastatin on knee cartilage volume in patients with symptomatic knee osteoarthritis: results from a randomised placebo-controlled trial.** *Arthritis & Rheumatology* 2021, *n/a*(n/a).
9. Wang Y, Jones G, Keen HI, Hill CL, Wluka AE, Kasza J, Teichtahl AJ, Antony B, O'Sullivan R, Cicuttini FM: **Methotrexate to treat hand osteoarthritis with synovitis (METHODS): an Australian, multisite, parallel-group, double-blind, randomised, placebo-controlled trial.** *The Lancet*.